# Exploring the Relationship between In-Context Learning and Instruction Tuning

## Abstract

In-Context Learning (ICL) and Instruction Tuning (IT) are two primary paradigms of adopting Large Language Models (LLMs) to downstream applications. However, they are significantly different. In ICL, a set of demonstrations are provided at inference time but the LLM's parameters are not updated. In IT, a set of demonstrations are used to tune LLM's parameters in training time but no demonstrations are used at inference time. Although a growing body of literature has explored ICL and IT, studies on these topics have largely been conducted in isolation, leading to a disconnect between these two paradigms. In this work, we explore the relationship between ICL and IT by examining how the hidden states of LLMs change in these two paradigms. Through carefully designed experiments conducted with LLaMA-2 (7B and 13B), we find that *ICL is implicit IT*. In other words, ICL changes an LLM's hidden states as if the demonstrations were used to instructionally tune the model. Furthermore, the convergence between ICL and IT is largely contingent upon several factors related to the provided demonstrations. Overall, this work offers a unique perspective to explore the connection between ICL and IT and sheds light on understanding the behaviors of LLM.

## 1 Introduction

Large language models (LLMs), such as ChatGPT [1], GPT-4 (OpenAI, 2023), PaLM (Chowdhery et al., 2022), and LLaMA-2 (Touvron et al., 2023), have significantly changed the paradigm of natural language processing and hold great potential for artificial general intelligence (Bubeck et al., 2023). In real-world applications, the success of deploying Large Language Models (LLMs) can largely be attributed to the effectiveness of two primary learning paradigms: 1) In-Context Learning (**ICL**) and 2) Instruction Tuning (**IT**). ICL, a paradigm introduced in the GPT-3 paper, involves utilizing a set of demonstrations are provided at inference time to guide the model's responses, but the model's parameters are not updated during this process. In contrast, IT refers to the process of further training LLMs on *input*, *output*, along with *instructions* in a supervised fashion. IT has been shown to be effective in enhancing an LLM's generalizability on unseen tasks (Longpre et al., 2023) and a viable strategy for LLM alignment (Taori et al., 2023; Zhou et al., 2023). Figure 1 illustrates ICL and IT using sentiment analysis as an example.

A growing body of literature has examined the mechanisms of ICL and IT, such as identifying the conditions under which ICL emerges in LLMs (Liu et al., 2021; Lu et al., 2021; Su et al., 2022; Wang et al., 2023; Chan et al., 2022; Xie et al., 2021), and determining how to design data and tasks for effective instruction tuning to enhance the zero-shot generalizability of LLMs (Longpre et al., 2023). However, while ICL and IT are two primary methods for enhancing the capabilities of LLMs, studies on ICL and IT have been conducted in isolation. This has led to a research question: What are the connections between ICL and IT, and in which way do they enhance an LLM's capability.

In this work, we examine the connection between ICL and IT via the hidden state of the input sequence's last token. In an autoregressive model, the hidden state of the input sequence's last token summarizes the information of the entire input sequence and determines the logit vector for the next word prediction. In the context of ICL and IT, three situations arise, each producing different hidden states. The first situation involves zero-shot learning for an LLM. In this case, the hidden state of the

---

[1] https://openai.com/chatgpt

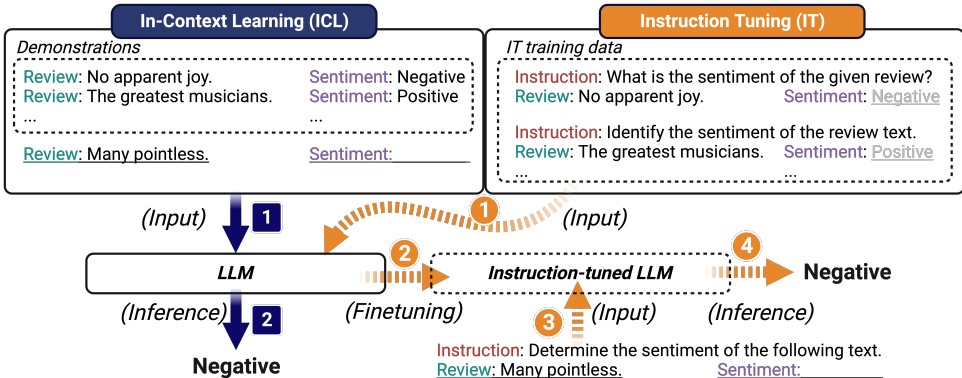

Figure 1: Illustrations for ICL and IT using sentiment analysis as an example. Through ICL, the LLM infers a *"Negative"* sentiment for *"Many pointless."* conditioned on a set of demonstrations (Left). In contrast, IT involves further tuning the LLM's parameters with the IT training data, and the tuned LLM is then used at inference time (Right).

last token in the input sequence is determined by the LLM, conditioned on the inference example. Since this is the basic case—where no demonstrations are provided and the LLM's parameters are not updated—we denote this as the anchor hidden state, $h_{anchor}$. The second situation is ICL, where demonstrations are provided to guide the LLM's response. Since ICL does not tune the LLM's parameters, the hidden state is determined by the LLM, conditioned on the provided demonstrations and the inference sample. We denote this hidden state as $h_{ICL}$. The third situation is IT, where demonstrations are used to tune the LLM's parameters, transforming the LLM into a tuned-LLM. Here, the hidden state is determined by the tuned-LLM, conditioned on the inference sample, and we denote this hidden state as $h_{IT}$. Comparing the similarity between $h_{anchor}$ and $h_{ICL}$ allows to quantify the effect of a demonstration in ICL, while comparing the similarity between $h_{anchor}$ and $h_{IT}$ allows to quantify the effect of IT with the demonstration. If a demonstration is effective for ICL and IT, we would observe a small similarity score because the demonstration gears the LLM to produce a guided (either through ICL or through tuning) response. Moreover, examining the similarity between $h_{ICL}$ and $h_{IT}$ allows us to directly quantify the extent to which ICL and IT on LLM converge, conditioned on the demonstrations. Figure 2 illustrates the analysis framework.

In the experiment, we select LLaMA-2 (7B) (Touvron et al., 2023) as the foundational LLM. We compile a demonstration dataset for sentiment analysis, consisting of tuples of <instruction, example, label>. Subsequently, we apply ICL and IT to LLaMA-2 using the same demonstration and examine the similarities between $h_{anchor}$, $h_{ICL}$, and $h_{IT}$. We repeat the experiment with variations in the wording of the instruction and demonstration examples. The results reveal a high similarity between $h_{ICL}$ and $h_{IT}$, while the similarity of these two hidden states with $h_{anchor}$ is low. This suggests that ICL and IT essentially guide the LLM to a similar status, although IT tunes the LLM's parameters while ICL does not. To further investigate, we vary the demonstrations used in ICL and IT and quantify the extent of similarity between ICL and IT conditioned on the demonstrations. For instance, we manipulate the number of demonstrations (from one-shot ICL to few-shot ICL), alter the semantic similarity between demonstration examples and inference examples, use a wrong label for the demonstration example, and employ different tasks as demonstrations. The results consistently support the finding that using a demonstration in ICL has a similar effect as using the demonstration to instructionally tune the LLM. In additional analyses examining the robustness of our findings, we change the inference task to a machine translation task and replace LLaMA-2 (7B) with LLaMA-2 (13B); the results remain consistent.

In summary, this work makes two contributions. First, we provide empirical evidence that ICL and IT are closely related. Although ICL does not alter model parameters—unlike IT—the instructions and demonstrations they employ drive the model towards convergent hidden states. Second, this study sheds light on how to design effective datasets and tasks for ICL and IT, potentially advancing the development and alignment of foundation models for downstream applications. We will make the experimental codes available for replication.

## 2    ANALYSIS FRAMEWORK

We illustrate our analysis framework in Figure 2, using sentiment analysis on reviews as an example. In this framework, we examine the impact of different demonstrations (zero-shot vs. few-shot ICL) and different paradigms (ICL vs. IT) on the model's hidden states separately. Although LLMs maintain hidden states for every input token, we primarily focus on the hidden states associated with the last input token of the sequence in this study. This focus is due to the hidden state of the last token of the last layer summarizing the information of the entire input sequence and determining the logit vector for the next word prediction.

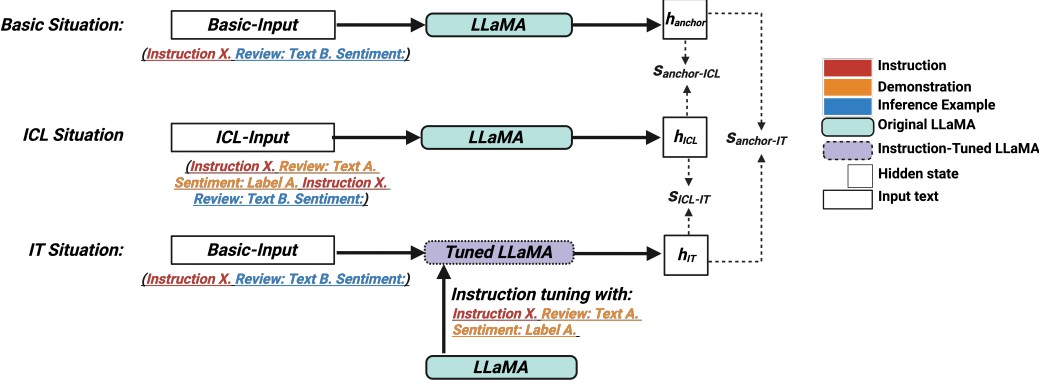

Figure 2: Analysis framework using sentiment analysis on reviews as an example. Our framework has variations by manipulating the demonstrations, changing the LLM, altering the input template, and adapting to different natural language tasks.

We denote instruction as $X$ (such as, *what is the sentiment of this review?*), demonstration as $A$=(Text A, Label A) (such as, *Review: This is a wonderful movie. Sentiment: Positive*), and inference text as $B$=(Text B) (such as, *Review: I like this movie.*). We then consider the following three situations.

**Basic situation.** This is the basic zero-shot learning setting where no demonstrations are provided to guide the model inference. In this situation, we concatenate instruction with the inference example (i.e., Instruction $X$ + Text $B$) and feed into an LLM. We collect the final hidden state of the last token of the input sequence, denoted as $h_{anchor}$.

**ICL situation.** In ICL, demonstrations, along with the inference example (i.e., Instruction $X$ + Text $A$ + Label $A$ + Text $B$), are provided as input to the LLM, which then directly infers the distribution of the last token. We collect the final hidden state of the last token of the input sequence, denoted as $h_{ICL}$. Comparing the similarity between $h_{anchor}$ and $h_{ICL}$ allows us to examine the effect of the provided demonstration. If the similarity is low, it indicates that the demonstration information are incorporated by the LLM so that the final hidden states are geared away.

**IT situation.** In IT, unlike the ICL situation where the demonstration is used as a part of input sequence, we instead use the demonstration (i.e., Instruction $X$ + Text $A$ + Label $A$) to instructionally tune the LLM, leading to a tuned LLM. We then send the inference example (i.e., Instruction $X$ + Text $B$) to the tuned LLM and obtain final hidden state of the last token, denoted as $h_{IT}$. Note that the input sequence to the final LLM are exactly the same (i.e., Instruction $X$ + Text $B$) in both the basic situation and the IT situation. The only difference is that the basic situation involves the vanilla LLM while the IT situation involves the instruction-tuned LLM. Therefore, by comparing $h_{anchor}$ with $h_{IT}$, we can quantify the effect of IT with the demonstration.

Since the same demonstration is used in both ICL and IT, we can precisely quantify the effect of the demonstration. By varying the provided demonstrations, we can also determine the extent to which ICL is related to IT, conditioned on the demonstrations. In the analysis, we further denote $s_{anchor-ICL}$ as the similarity between $h_{anchor}$ and $h_{ICL}$, and denote $s_{anchor-IT}$ as the similarity between $h_{anchor}$ and $h_{IT}$. We also measure the similarity between $h_{ICL}$ and $h_{IT}$, denoted as $s_{ICL-IT}$, which quantifies the extent to which ICL and IT converge. If the $s_{ICL-IT}$ is very

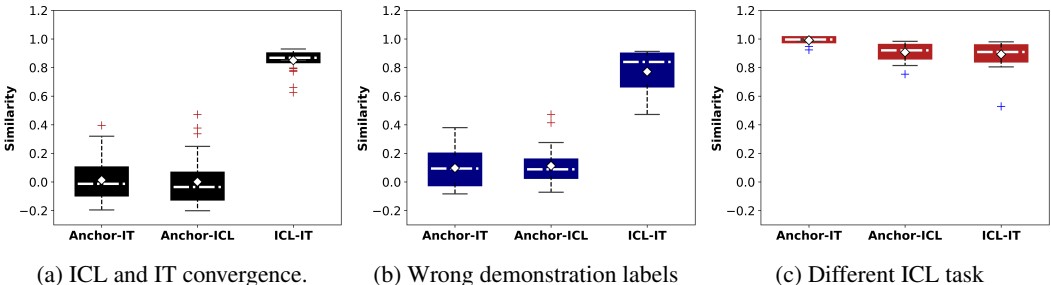

(a) ICL and IT convergence.  (b) Wrong demonstration labels  (c) Different ICL task

Figure 3: Similarities between different hidden states. We use the box plots to show the distribution of scores in the repeated experiments.

high, it indicates ICL and IT guide the model status towards the same direction although the model parameters are not updated in ICL but tuned in IT.

## 3 EXPERIMENTS

### 3.1 EXPERIMENT SETUP

**Datasets:** In the experiment, we use the SST2 for sentiment analysis (Socher et al., 2013) and EN-CS of WMT16 for English-Czech translation (Bojar et al., 2016). For each of the tasks, we manually craft a pool of instructions and randomly choose instruction in the repeated experiment, alleviating the concern that the experiment results are driven by a specific instruction. Instructions used for each task are presented in Appendix A.

**LLMs:** We use LLaMA-2-base as the foundation model (Touvron et al., 2023), including 7B (32 layers with a hidden size of 4,096) and 13B (40 layers with a hidden size of 5,120). We download the models following the instructions from Meta AI [2], and implement them using the `transformers` library [3].

**Instruction tuning:** We use the LoRA technique (Hu et al., 2021) to instruction-tune the LLaMA-2 model due to its efficiency. Specifically, we target modules $Q$ and $V$, and use a dropout probability 0.05, learning rate 1e-4, scaling factor 32, and a rank of 8. We use AdamW optimizer (Loshchilov & Hutter, 2017). Without further specification, we tune the model with 10 epochs and use *bf16* precision.

**Repeated experiment:** In the following analysis, we randomly choose an instruction, a demonstration and an inference example from the dataset for ICL and IT. We repeat the procedure for 30 runs with different random seeds.

### 3.2 EMPIRICAL FINDINGS

We present the empirical findings as follows.

**ICL and IT convergence: In-Context Learning (ICL) and Instruction Tuning (IT) result in a converged model state.** We present the hidden state similarities in Figure 3a. Firstly, we observe that the similarity between $h_{anchor}$ and either $h_{ICL}$ or $h_{IT}$ is almost zero, indicating that the model undergoes significant changes in its hidden representations when exposed to in-context demonstrations or when tuned by the demonstrations. Furthermore, the high similarity between $h_{ICL}$ and $h_{IT}$ (approximately 0.9) demonstrates that the model is indeed oriented toward a similar state in ICL and IT. This provides a first evidence that ICL is implicit IT.

**Demonstration-inference similarity: The convergence between ICL and IT is positively correlated with the semantic similarity between the demonstration and the inference example.** We further investigate how the semantic similarity between the demonstration (i.e., Text $A$ in Figure

---

[2] https://github.com/facebookresearch/llama
[3] https://github.com/huggingface/transformers

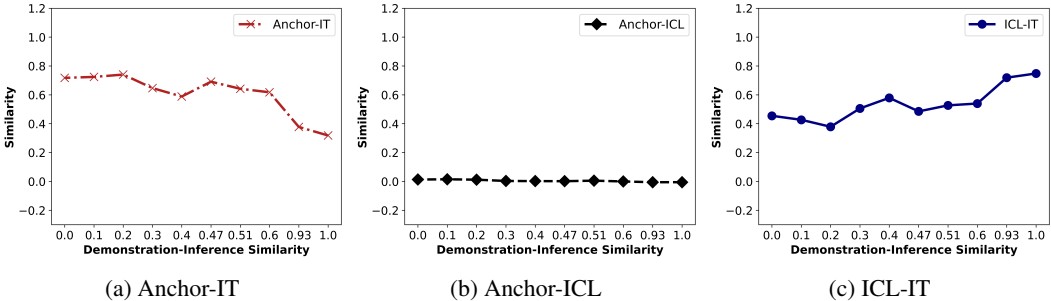

| (a) Anchor-IT | (b) Anchor-ICL | (c) ICL-IT |

Figure 4: Averaged hidden state similarities across demonstration-inference similarity levels.

2) and the inference example (i.e., Text $B$) affects the ICL-IT convergence. To do this, we use a sentence-transformer model "all-MiniLM-L6-v2" [4] to measure the demonstration-inference similarity (Reimers & Gurevych, 2019). We consider 10 levels of similarity ranging from 0 to 1. For each inference example, we identify demonstrations in the dataset that fall within a specific similarity range. In each repeated experiment involving different similarity levels, we randomize the input but use the same set of inference examples across these cases to facilitate a fair comparison. The results are shown in Figure 4. Clearly, the similarity between ICL and IT increases as the similarity between the demonstration and the inference example increases (Figure 4c). A possible explanation is that a demonstration that is more similar to the inference example can better enhance the model's ICL ability and is also more helpful for IT, resulting in higher convergence. It is worth noting that the range of the degree of convergence between ICL and IT is quite large, ranging from around 0.4 when they are entirely different (demonstration-inference similarity is 0) to 0.8 when they are exactly the same (demonstration-inference similarity is 1).

In contrast, the similarity between $h_{anchor}$ and $h_{IT}$ exhibits an opposite trend, as shown in Figure 4a, suggesting that a demonstration that is more similar to the inference example can change the model's state to a greater extent. This finding aligns with prior literature, which has demonstrated that instruction tuning with similar examples is more effective (Gudibande et al., 2023). Put it another way, fine-tuning the model with semantically different examples does not substantially alter the model's inference capability.

Interestingly, we observe that the similarity between $h_{anchor}$ and $h_{ICL}$ remains consistently low, regardless of the demonstration-inference similarity, as illustrated in Figure 4b. This suggests that incorporating demonstrations into the ICL input can consistently and significantly impact the model's inference. Previous studies on ICL have indicated that higher demonstration-inference similarity leads to improved inference accuracy. It's important to emphasize that Figure 4b does not contradict this finding, as it measures the similarity between $h_{anchor}$ and $h_{ICL}$.

**Number of demonstrations: The convergence between ICL and IT increases as the number of demonstration increases.** In the previous analysis, we used a single demonstration in ICL and IT. In this experiment, we vary the number of demonstrations (i.e., few-shot learning) in ICL and IT. Specifically, we consider 1-shot, 2-shot, 5-shot, and 10-shot scenarios. To ensure a fair assessment, we maintain consistent parameters update times and instruction-tune the model with 10, 5, 2, and 1 epoch(s), respectively. For each repeated experiment in the various few-shot cases, we randomize the input but use the same set of inference examples across these cases to enable a fair comparison.

We present the results in Figure 5. We observe a clear increasing trend in the convergence between ICL and IT as we incorporate more demonstrations. This trend is intuitive since ICL with multiple demonstrations (i.e., few-shot learning) can help the model discover patterns in the context and quickly adapt to the task. Similarly, IT using more examples related to the same task can better tune the model for that specific task, leading to a higher level of convergence between ICL and IT.

**Wrong label: Demonstration with wrong label slightly affects the ICL-IT convergence.** Prior studies in ICL have shown that the correctness of demonstration's label does not matter much and only the task format is important for ICL (Min et al., 2022). Therefore, it motivates us to examine

---

[4] https://www.sbert.net

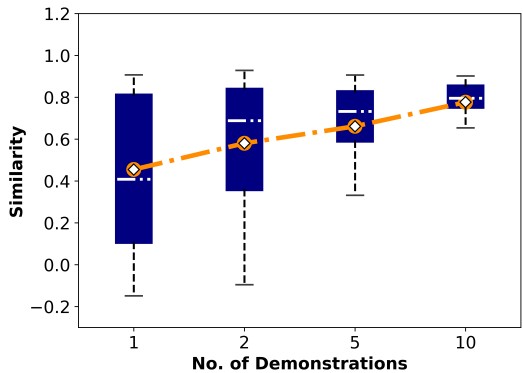

Figure 5: ICL-IT convergence across different numbers of demonstrations.

how the label correctness affects the ICL-IT convergence. In this experiment, we reverse the labels of demonstrations (e.g., changing "Positive" to "Negative"), and conduct the ICL and IT procedure again. The results are shown in Figure 3b.

Interestingly, we find that while ICL and IT still exhibit a high level of convergence, the degree is slightly lower than its counterpart when using correct labels as compared to Figure 3a. Besides, the variation of the degree of ICL-IT convergence significantly increases, as evidenced by the larger interquartile range and longer whiskers of the box plot.

As a sanity check, we examine if using wrong labels to do IT hurts the model performance, and present the results in Figure 6. Surprisingly, although we do observe a performance drop, the decrease is not statistically significant, which appears to be well aligned with previous observations in (Kung & Peng, 2023).

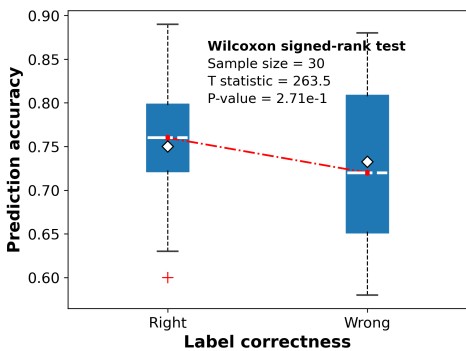

Figure 6: Prediction accuracies of using wrong demonstration labels vs. right. We perform one-tailed Wilcoxon signed-rank test, and the null hypothesis is the difference between paired observations (right-wrong) is greater than zero.

**Different task: Different demonstration task would not affect the ICL-IT convergence.** In the previous experiments, the demonstration task and the inference task are the same (i.e., sentiment analysis). This experiment differs in that we change the demonstration task to machine translation using the EN-CS subset of WMT16 translating English to Czech [5], but the sentiment analysis remains the inference task. We present the results in Figure 3c. Clearly, the high level of convergence in similarities between ICL-IT, Anchor-ICL, and Anchor-IT indicates that the demonstrations involving the machine translation task do not impact the model's inference capability for the sentiment analysis task.

**Intermediate layers: The convergence between ICL and IT starts to increase at later layers.** In this experiment, we examine the hidden states of the last token of the input sequence in all layers of

---

[5]We use the following template: *"Instruction X. English: English A. Czech: Czech A. Instruction X. English: English B. Czech:"*.

the LLM. The results are shown in Figure 7. Interestingly, we observe an U shape across different layers. The high similarity between ICL and IT in the lower layer is primarily due to the fact that the hidden states are all similar to the anchor hidden states, meaning they are not significantly impacted by the demonstrations. The LLM's intermediate layers are gradually influenced by the demonstrations, resulting in the low similarity between ICL and IT in the middle layers. Eventually, as the input approaches the higher layers that are closer to the final output, the hidden states of ICL and IT start to converge.

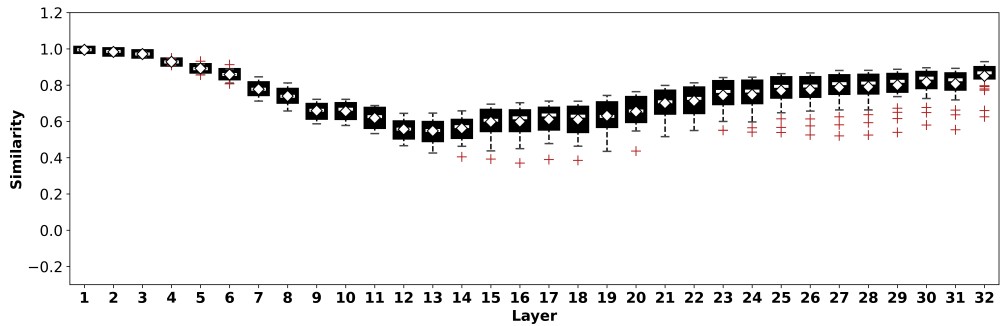

Figure 7: ICL-IT convergence scores of all layers.

# 4 ADDITIONAL ANALYSIS

## 4.1 LLAMA-2-13B

In this study, we examine if ICL and IT still converges in a larger LLM. We choose LLaMA-2-13B as the foundation model and repeat the same analysis procedure to quantify the similarity between Anchor-IT, Anchor-ICL and ICL-IT. The results are shown in Figure 8a, indicating that ICL-IT convergence remains high. However, Anchor-IT and Anchor-ICL also achieve a high level of convergence, indicating that larger model is more capable of understanding the task even without any demonstrations provided (note that in the basic situation, an instruction is provided which could provide sufficient information for the larger LLM to do zero-shot learning).

## 4.2 SUPERVISED LEARNING

Instruction tuning differs from classic supervised learning in that the former employs additional instructions to enhance an LLM's generalizability, while supervised learning typically teaches the LLM to specialize in a specific task.

To further understand the role of instructions in IT, we conduct classic supervised learning for the LLM. In this setup, we remove Instruction $X$ from the training input and solely use task examples to fine-tune the LLM. We denote this supervised situation as SL. We repeat the same analysis procedure and measure the similarity between Anchor-SL, Anchor-ICL, and ICL-SL. We present the results in Figure 8b. Clearly, while the convergence between ICL and SL still exists, the convergence score is significantly lower than that of its IT counterparts, as shown in Figure 3a. This observation underscores the critical role of instructions in driving the convergence between ICL and IT in LLMs' hidden states.

## 4.3 UNDERSTANDING INSTRUCTION TUNING FROM IN-CONTEXT LEARNING

Evidences discussed above suggest that ICL is essentially doing IT via demonstrations. In this section, we aim to understand IT through the lens of ICL. Specifically, instead of focusing on the hidden states, we calculate the change of *per token loss* of the LLM. We define *per token loss* as the cross-entropy loss between each output token and the corresponding ground truth token in a sequence (Olsson et al., 2022). We illustrate the procedures of the experiment in Figure 9. The major steps are as follows. Firstly, we randomly sample an instruction $X$ and an example $A$. We then construct the input using the template shown in Figure 2 as: *"Instruction $X$. Review: Text*

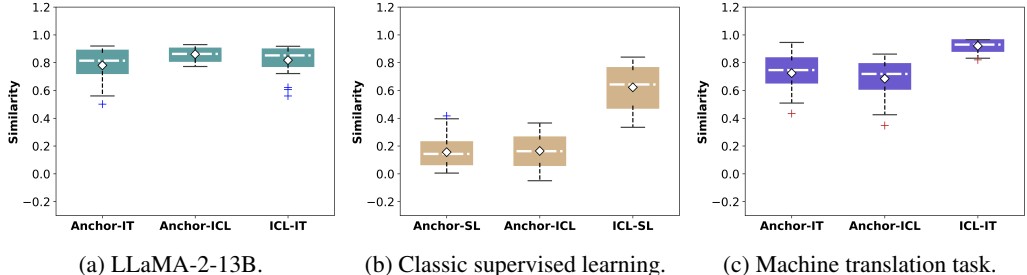

(a) LLaMA-2-13B.  (b) Classic supervised learning.  (c) Machine translation task.

Figure 8: Similarities between different hidden states (additional analysis).

*A. Sentiment: Label A.".* Next, we send the input to LLaMA-2-7B and collect the per token loss. After that, we instruction-tune the language model using this example. After tuning, we send the same input again to the tuned model and collect the per token loss. We then calculate the loss decrease for each token and average the per token loss decrease by token's identity (i.e, "Instruction" or "Example"). We conduct 30 independent experiments using different seed values. The results are shown in Figure 10. Clearly, we observe a more significant loss decrease for the "Example" component compared to the "Instruction" component, suggesting the tuned model is more likely to reproduce task relevant examples given an instruction. In other words, the instruction is somehow substituted by the examples it associates at inference time, leading to a similar input format as ICL.

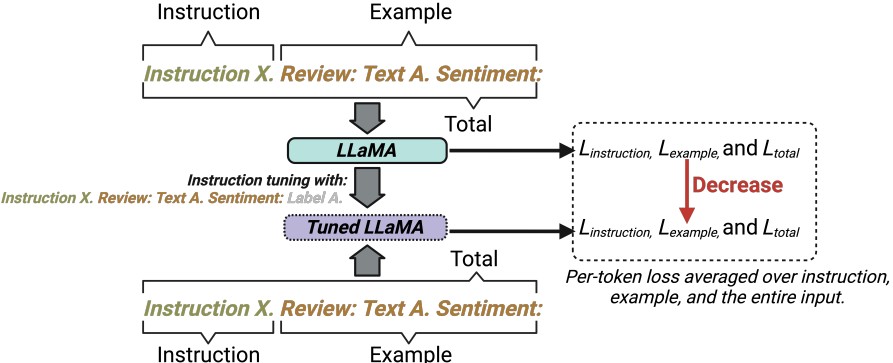

Figure 9: Illustration: The decreased loss indicates instruction can help the model associate relevant examples at inference time.

## 4.4 ROBUSTNESS CHECK: MACHINE TRANSLATION

As a robustness check, we replace the sentiment analysis task (a natural language inference task) with the machine translation task (a natural language generation task), and conduct the same procedure to examine if the connection between ICL and IT still holds. We choose a machine translation task that translates English text into Czech using the EN-CS subset of WMT16 dataset (Bojar et al., 2016). We present the results in Figure 8c. It is interesting to note that the similarity between ICL and IT is remarkably high. Recall that the input examples for ICL and IT are very different. The substantial similarity between ICL and IT supports the earlier findings that ICL, when using demonstrations, significantly alters an LLM's inference capability, akin to how demonstrations are used to fine-tune the LLM.

Unlike sentiment analysis, where the similarity between Anchor-IT and Anchor-ICL is as low as zero, the similarity is higher in the machine translation task. However, a statistical test reveals that the similarity between ICL and IT is statistically greater than that between Anchor-IT and Anchor-ICL[6]. This rules out the possibility that all three hidden states are very similar to each other.

---

[6]We conducted a one-tailed Wilcoxon signed-rank test between each pair of them. The p-value is $2.79e - 9$ for Anchor-IT and ICL-IT, and $9.31e - 10$ for Anchor-ICL and ICL-IT. The sample size is 30, and the

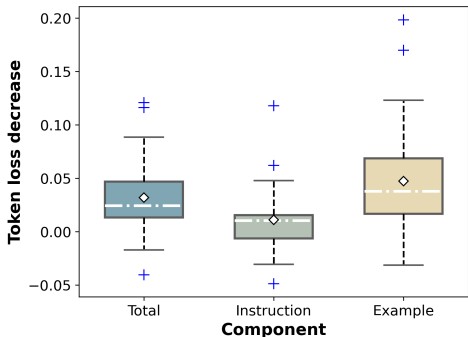

Figure 10: Per-token loss decrease due to instruction tuning.

## 5 RELATED WORK

**In-Context Learning (ICL)** is a phenomenon emerged in large language models (Brown et al., 2020). A growing body of literature has investigated the ICL phenomenon in LLMs. Some studies have focused on identifying the conditions under which ICL emerges in LLMs, predominantly by finding good demonstrations (Liu et al., 2021; Lu et al., 2021; Su et al., 2022; Wang et al., 2023) and identifying pre-training data distributions that can lead to the emergence of ICL (Chan et al., 2022; Xie et al., 2021). Another line of research aims to explain ICL through building the relationship with the model training stage (Akyürek et al., 2022; Dai et al., 2022; Li et al., 2023; Von Oswald et al., 2023). For instance, Akyürek et al. (2022) find ICL implicitly updates smaller models encoded in the activations. Olsson et al. (2022) provide evidence that the so-called "induction heads" contribute to the majority of the ICL behaviors in LLMs.

Our work differs from existing studies in two ways. First, we attempt to understand ICL by investigating its connection with IT, which is new and opens up the possibilities for harnessing the complementary knowledge of ICL and IT. Second, we empirically study off-the-shelf LLMs with much more complex model structures (LLaMA-2 7B and 13B), whereas most prior works conduct experiments using more simplified models (Li et al., 2023).

**Instruction Tuning (IT)** is an efficient technique to adapt LLMs to downstream tasks by further tuning the model on (*"input"*, *"output"*) pairs with instructions in a supervised manner. The intuition behind IT is to bridge the gap between the language modeling objective in pre-training and the users' objective in downstream tasks, such that the model can follow the instructions from users. The effectiveness of IT is well-demonstrated by a variety of instruction-tuned LLMs, with representatives such as InstructGPT (Ouyang et al., 2022), Alpaca (Taori et al., 2023), Flan-T5 (Longpre et al., 2023), and Vicuna [7]. A growing body of literature focuses on designing tasks and datasets for effective instruction tuning. For example, LIMA (Zhou et al., 2023) shows that a small set of high-quality instruction datasets is sufficient for foundation model alignment. Our work aims to provide empirical evidence to further understand IT, through the lens of its connection with ICL.

## 6 CONCLUSIONS

In this work, we explore the connection between in-context learning (ICL) and instruction tuning (IT). Through carefully designed experiments, we provide strong evidences suggesting ICL is implicitly IT. In other words, ICL changes an LLM's hidden states as if the demonstrations were used in IT. This finding sheds light on the behaviors of two very different learning paradigms of LLM (ICL vs. IT), potentially benefiting the development and alignment of foundation LLMs to downstream real-world applications.

---

null hypothesis is that the difference between paired observations ($s_{ICL-IT} - s_{anchor-IT}$ and $s_{ICL-IT} - s_{anchor-ICL}$) is greater than zero.

[7] https://lmsys.org/blog/2023-03-30-vicuna/

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

## A    INSTRUCTION SETS

What is the sentiment of the movie review below? Is it negative or positive?
Determine whether the sentiment expressed in this movie review is negative or positive:
Identify whether this movie review contains negative or positive opinions.
Classify whether this movie review conveys negative or positive opinions.
Rate whether the viewpoint on the costumes is more negative or positive.
Based on the review content, would you say the sentiment is negative or positive?
Analyze the sentiment expressed in this movie review. Is it positive or negative?
Identify negative or positive of the content.
Evaluate the sentiment of this movie critique. Is it negative or positive?
Determine the sentiment conveyed in this movie review. Is it negative or positive?
Classify the overall sentiment of this movie review as negative or positive.
Determine if the tone of this movie review is negative or positive.
Assess if the tone of this movie review is negative or positive.
Detect whether this movie review contains negative or positive sentiment.
Determine whether this movie review expresses negative or positive sentiment.
Identify whether the sentiment expressed in this movie review is negative or positive.
Distinguish whether the evaluation in this movie review is negative or positive.Provide your answer as either negative or positive:
Infer whether the tone of this movie review is negative or positive.
Grade if the perspective in this movie review is negative or positive.Provide your answer as either negative or positive:
What's the emotional tone of this movie review? Would you describe it as negative or positive?
Infer whether this movie review expresses negative or positive emotion.
Estimate if the analysis in this movie review is negative or positive.Provide your answer as either negative or positive:
Determine whether the opinions in this movie review are negative or positive.
Identify the sentiment of the following movie review text. Is it negative or positive?
Assess the sentiment expressed in the following movie review. Is it positive or negative?
Determine the sentiment expressed in this movie review. Negative or positive?

Table 1: Instructions for sentiment analysis.

Can you express this English phrase in Czech?
Can you present this English sentence in Czech?
Please make this English sentence into a Czech sentence.
Please convert this English text into Czech.
Help me interpret this English phrase in Czech.
Translate this English sentence into Czech.
Please provide a Czech translation for this English sentence.
I need your help to change this English sentence into Czech.
Could you help convert this English phrase into Czech?
Could you translate this English text into Czech?
Please, translate the following English sentence into Czech.
Rephrase this English sentence in Czech for me, please.
Please give me the Czech version of this English sentence.
Can you assist in translating this English sentence into Czech?
Can you change this English sentence into Czech?
How would you say this English sentence in Czech?
Please convert this English phrase into Czech.
Can you convert this English sentence into Czech, please?
Please interpret this English sentence into Czech for me.
Please provide a Czech version of this English sentence.
Can you give me a Czech translation of this English text?
Could you kindly convert this English text into Czech?
Could you rewrite this English phrase in Czech?
I require this English sentence to be translated to Czech.
I need this English phrase translated to Czech.
Translate this English content into the Czech language, please.
Translate this English phrase into Czech for me, please.
Can you provide a Czech interpretation of this English sentence?
Can you render this in the Czech language, please?
Can you transcribe this English text into Czech?

Table 2: Instructions for machine translation.

