# OpenReview forum: "Exploring the Relationship between In-Context Learning and Instruction Tuning"
_ICLR.cc/2024/Conference — ICLR 2024 Conference Withdrawn Submission_

### Official Review · Reviewer_g5Tk · 2023-10-30

**Soundness:** 2 fair
**Presentation:** 3 good
**Contribution:** 2 fair
**Rating:** 5
**Confidence:** 4

**Summary:**

This paper investigates the previously unexplored relationship between in-context learning (ICL) and instruction tuning (IT) in natural language processing (NLP). The significance of this study lies in its examination of the potential linkage between these two distinct adaptation methods for large language models (LLMs). Understanding this connection is crucial as it may allow for the combined use of ICL and IT, leading to more robust and generalizable models. To explore this relationship, the authors focus on how the hidden states of LLMs are altered under each paradigm. They compare similarity metrics of the last token's hidden state in input sequences under both methods. Interestingly, the findings indicate that in-context learning modifies the hidden states of LLMs in a manner similar to how intentional model tuning would. The evaluation of this relationship predominantly centers on sentiment analysis, with a supplementary analysis conducted for translation tasks.

**Strengths:**

**Intriguing Research Question:**
The paper addresses an interesting and relevant question within the field of NLP: exploring the relationship between in-context learning (ICL) and instruction tuning (IT). This inquiry is particularly pertinent given the growing complexity and capabilities of large language models.

**Well-Designed Experimental Framework:**
The careful and methodical design of the experimental setup is a notable strength.

**Rigorous Numerical Comparison:**
The numerical analysis conducted in this paper is thorough and reliable. The use of different seeds and the reporting of standard deviations in the results showcase a commendable level of rigor and transparency.

**Weaknesses:**

**Generalizability of Findings:**
The paper's primary limitation arises from the constrained generalizability of its findings. The focus on sentiment analysis as the primary evaluation task seems inadequate for thoroughly examining the relationship between in-context learning (ICL) and instruction tuning (IT). This limitation is particularly evident in Figure 8, where scaling the model size from 7B to 13B parameters doesn't reveal a clear relationship between ICL and IT. A more diverse range of complex tasks might be necessary to understand these paradigms' interplay more comprehensively.

**Clarity and Structure of the Content:**
Certain sections of the paper, such as those discussing Figures 3.c and 8.c, lack a clear and cohesive structure. Ideally, each section should be framed with a set motivation, detailed experimentation, justification, and a conclusive statement. These elements are noticeably missing or inadequately presented in the mentioned sections, leaving the reader without a clear understanding or a conclusive message, undermining the paper’s coherence and impact.

**Clarification on Specific Statements:**
In section 4.3, the statement, "In other words, the instruction is somehow substituted by the examples it associates at inference time, leading to a similar input format as ICL," is unclear and inadequately explained. More elaborate experimental details, underlying logic, and clarification would be beneficial. This clarification is vital to understand how the models’ learning dynamics operate under varying conditions.

**Unclear Second Contribution:**
The paper claims in its introduction that a secondary contribution is shedding light on designing effective datasets and tasks for ICL and IT, aiding the development of foundational models. However, upon reviewing the document, it becomes apparent that there is a notable lack of clear, direct statements or findings supporting this claim. Explicit elaboration on this contribution, with concrete recommendations or insights, would substantially strengthen this aspect of the paper.

**Questions:**

**Generalizability of Findings:**

The primary concern regarding the paper lies in the generalizability of its findings, particularly pertaining to the central hypothesis that "ICL is implicit IT." The current experimental setup, primarily focusing on sentiment analysis, seems insufficient for a robust examination of this hypothesis. The simplicity of the sentiment analysis task may not adequately capture the nuanced dynamics between ICL and IT.

To address this, I would strongly recommend that the authors incorporate more complex tasks, such as reasoning or question answering, into their experimental design. These tasks, by nature, require a more sophisticated understanding and application of knowledge, potentially making the similarities or differences between ICL and IT more pronounced and observable. I am insisting on this as more complex tasks indeed need ICL or IT for better performance.

P.S.: I would raise my score from weak reject to weak accept if the authors address the above concern.

---

### Official Review · Reviewer_i92c · 2023-10-30

**Soundness:** 2 fair
**Presentation:** 3 good
**Contribution:** 2 fair
**Rating:** 3
**Confidence:** 4

**Summary:**

In this study,  it explores  the relationship between in-context learning (ICL) and instruction tuning (IT). Their meticulously crafted experiments offer compelling evidence that ICL essentially functions as IT. ICL modifies an LLM's hidden states in a manner similar to how demonstrations impact IT. This insight provides clarity on how two distinct LLM learning approaches (ICL and IT) operate, which could be valuable for refining and aligning foundational LLMs for practical applications.

**Strengths:**

The paper is easy to follow and well written, the observations look like novel to me.

**Weaknesses:**

1. The author's method of measuring similarity between h_{anchor} and h_{IT} is a bit confused to me. As I comprehend it, the author compares the final token embeddings to determine cosine similarity between different techniques, for example between anchor and IT. However, I'm skeptical about using the last token embedding to assess the impact of few-shot learning or IT.
2. Additionally, the paper utilizes sentiment analysis datasets. I'm not convinced that these are suitable text-to-text datasets to substantiate the paper's conclusions. Given that the outputs are merely "positive" or "negative," this seems overly simplistic for large language models, especially in the context of few-shot or instruction tuning. Therefore, my takeaway is that sentiment analysis might be relatively straightforward for few-shot or instruction tuning setups. I'm not convinced by the paper's claim "provide strong evidences suggesting ICL is implicitly IT." It seems more likely that both ICL and IT can easily manage this task, leading to similar embeddings behavior. I suggest the author should try more datasets from other few-show paper, for example datasets in Chain-of-Thoughts paper.
3. I'd be curious to see task performance reported under these different conditions.

**Questions:**

Please see the weakness part.

---

### Official Review · Reviewer_szc9 · 2023-11-02

**Soundness:** 4 excellent
**Presentation:** 4 excellent
**Contribution:** 3 good
**Rating:** 8
**Confidence:** 4

**Summary:**

The paper explores the relationship between in-context learning and instruction tuning by analyzing the changes of internal representations of the last token driven by both in-context learning and instruction tuning. Through experimentation on LLAMA-2, the paper finds convergence between hidden states of the last token in the instruction-tuned model and in the model leveraging in-context learning with demonstrations. The paper concludes that in-context learning implicitly emulates instruction tuning.

**Strengths:**

Strengths:
* Innovative and important research question
* Carefully designed multifaceted experiments exploring the similarity of hidden states of the last token between ICL and IT with up to 30 times resampled input, robust results
* The main finding of ICL being implicit IT is very important for understanding the inner workings of LLMs and for further advances in the field
* The convergence between last token representations of IT and ICL phenomenon with the increase of the number of demonstrations presented in Figure 5 is very insightful
* Interesting experiments on the evolution of the ICL-IT similarity throughout the layers of the model

Overall, excellent and very thought-provoking paper, I enjoyed reading it.

**Weaknesses:**

Weaknesses and Questions:
* While similarity of ICL with anchor is shown to be low, that should depend on the number of ICL demonstration — it would be interesting to see some progression of ICL-anchor similarity depending on the number of demonstrations, maybe Figure 5 could be augmented with this plot.
* The introduction mentions that this study sheds light on how to design effective datasets and tasks for ICL and IT, potentially advancing
the development and alignment of foundation models for downstream applications. From the perspective of designing effective datasets, what would be the main takeaway of the paper and the most actionable insight?
* Are the observed effects dependent on LoRA tuning? Would they still hold if full instruction tuning was done?
* Why does similarity in Figure 4c not reach the 0.9 similarity as  in Figure 3a?
* While the design choice of tracking only the representation of the last token makes total sense, out of curiosity, do you have experiments with similar analysis for representations of other tokens?
* Figure 3c is really interesting. Does it mean that the internal representations aren’t affected by the in-context demonstrations as much as they are by the inference task?
  * Does the above mean that the model only takes into account demonstrations if they align with the inference task, otherwise it ignores them? How does this mechanism work?
  * Can Figure 3c be thought of as an extreme case of dissimilarity between the demonstration and inference examples as presented in Figure 4, when both the examples and the tasks are different? In Figure 4 there is no Anchor-ICL similarity while in Figure 3c there is high Anchor-ICL similarity, how would you explain the difference?

* Please explain this point more, to me it does not directly follow from 3c: *“Clearly, the high level of convergence in similarities between ICL-IT, Anchor-ICL, and Anchor-IT indicates that the demonstrations involving the machine translation task do not impact the model’s inference capability for the sentiment analysis task.”*
    * How do we know from Figure 3c that there is no degradation of sentiment analysis inference performance because of showing demonstrations of a different task? Potentially the performance could drop even with high similarity between the hidden states.

* Does the ICL-IT convergence phenomenon depend on the difficulty of the task? Have you tried it with other tasks, except sentiment analysis and translation?

**Questions:**

Please see the previous section for both weaknesses and questions.

**Details Of Ethics Concerns:**

no ethics concerns

---

### Official Review · Reviewer_Z5vk · 2023-11-08

**Soundness:** 3 good
**Presentation:** 3 good
**Contribution:** 3 good
**Rating:** 5
**Confidence:** 3

**Summary:**

This paper focuses on the convergence of in-context learning (ICL) and instruction tuning (IT) within large language models (LLMs). The experiments are set up to analyze the impact of different demonstrations and paradigms on the model's hidden states, with a focus on sentiment analysis and English-Czech translation tasks. The paper includes a detailed analysis framework and presents findings on how the correctness of demonstration labels and the task differences affect ICL-IT convergence.

**Strengths:**

1. The paper's approach to understanding the convergence of ICL and IT is commendable. It addresses a critical aspect of LLMs' ability to adapt and learn from instructions, which is central to their real-world applicability.

2. The findings that incorrect labels do not significantly impact performance and that task differences do not affect ICL-IT convergence are intriguing. They suggest a certain robustness in LLMs that warrants further investigation.

**Weaknesses:**

The paper appears to be a solid contribution to the field, it provides interesting insights and solid experiments. However, certain limitations should are also very transparent.

1. The paper's focus on sentiment analysis and English-Czech translation tasks and the use of specific instruction types could indeed limit the generalizability of the findings. Sentiment analysis tasks have their own unique characteristics and may not represent the full spectrum of tasks that LLMs are expected to perform. If the model's performance is optimized for these tasks, it may not perform as well on other types of tasks, such as reasoning or domain-specific knowledge tasks.

2. The use of the LLaMA-2 7B/13B model and LoRA for fine-tuning presents a narrow slice of the possible model architectures and fine-tuning methods available. While LoRA is known for its efficiency, there may be trade-offs in terms of how well it generalizes across different tasks or scales with different model sizes. The findings might not be applicable to models that are fine-tuned using other methods or have different architectural designs.

3. The paper's methodology involves the use of demonstrations in both ICL and IT. If the demonstrations are not diverse enough or too tailored to specific tasks, there's a risk that the model could overfit to these examples. This overfitting could manifest as an inability to perform well on tasks that deviate from the structure or content of the training demonstrations.

4. The analysis seems to focus on the similarity between demonstrations and inference examples. While this is an important aspect, it may not fully account for the complexity of real-world applications where the relationship between training examples and real-world data can be much more nuanced.

5. The paper's experiments include English-Czech translation tasks, which is a step towards cross-lingual generalizability. However, why only did experiments on certain type of translation tasks? it's not clear how well the findings would hold across a wider array of languages, especially those that are typologically different from English or have less representation in training data.

**Questions:**

My questions have been provided in above Weaknesses section.

---

### Official Review · Reviewer_GTLB · 2023-11-09

**Soundness:** 2 fair
**Presentation:** 3 good
**Contribution:** 3 good
**Rating:** 3
**Confidence:** 4

**Summary:**

This paper shows the connection between in-context learning (ICL) and instruction tuning (IT) for large language models (LLMs) through novel empirical analysis. By quantifying hidden state similarity, the work provides evidence that ICL and IT guide LLMs to converge to similar hidden states despite their different mechanisms. The results offer new insights into LLM behavior under varying inference schemes. Overall, the paper makes a contribution in uncovering the close relationship between two dominant LLM paradigms, enhancing our understanding of few-shot tuning approaches commonly used in practice.

**Strengths:**

Originality:
Examines hidden states to understand ICL and IT, an original perspective distinct from prior work.
Analyzes real-world LLMs rather than simplified models used in related studies.

Quality:
Technically rigorous methodology with nice experimental design.
Replicable experiments on real LLMs.

Clarity:
Clearly motivates exploring relationship between ICL and IT and frames an interesting research question.
Nice presentation of analysis framework and transparent description of experiments.

Significance:
Improves understanding of how models adapt during inference without parameter updates.

**Weaknesses:**

Similarity Measure Ambiguity
The similarity metric for comparing hidden states is unspecified. The measure should be clearly described to interpret meanings.

Limited Evaluation
Only two datasets are used for evaluation. Testing on more diverse tasks and datasets would demonstrate robustness. Examples: summarization, QA.

Unclear Practical Impact
While promising implications for dataset design, the paper lacks any clear guidelines for creating effective ICL demonstrations and IT data. Providing specific, applied best practices is needed.

Narrow Scope
The scope focuses solely on hidden state similarity. Expanding to other metrics and tasks would strengthen conclusions. For example, evaluating performance, analyzing attention.

Limited Generalizability
Unclear if findings generalize beyond the specific LLM setup i.e LORA, LORA target models (Q and V) and BF16. Experiments on more model types would show broader applicability.

Lack of Ablation Studies
The paper does not ablate model components or mechanisms to isolate their impact. Ablating instructions, tuning approaches, model sizes, etc could provide more thorough understanding about necessities. Specially for an empirical paper this is essential.

**Questions:**

Please address the weaknesses.